# The Prognostic Role of Automated Office Blood Pressure Measurement in Hypertensive Patients with Chronic Kidney Disease

**DOI:** 10.3390/healthcare11101360

**Published:** 2023-05-09

**Authors:** Konstantinos Psounis, Emmanuel Andreadis, Theodora Oikonomaki, Stefanos Roumeliotis, Vasileios Margellos, Elias Thodis, Ploumis Passadakis, Stylianos Panagoutsos

**Affiliations:** 1Department of Hemodialysis, Athens Medical Group, Dafni Clinic, 17237 Athens, Greece; 2Internal Medicine, Athens Medical Group, Psychiko Clinic, 11525 Athens, Greece; 3Department of Nephrology “Antonios Billis”, Evangelismos General Hospital, 10676 Athens, Greece; 4Division of Nephrology and Hypertension, 1st Department of Internal Medicine, AHEPA Hospital, School of Medicine, Aristotle University of Thessaloniki, 54636 Thessaloniki, Greece; 5Department of Nephrology, University Hospital of Alexandroupoli, 68100 Alexandroupoli, Greece

**Keywords:** automated office blood pressure, chronic kidney disease, hypertension, cardiovascular events, renal events

## Abstract

Background: The aim of this study was to evaluate the prognostic value of automated office blood pressure (AOBP) measurement in patients with hypertension and chronic kidney disease (CKD) stage 3–5 not on dialysis. Methods: At baseline, 140 patients were recruited, and blood pressure (BP) measurements with 3 different methods, namely, office blood pressure (OBP), AOBP, and ambulatory blood pressure measurement (ABPM), were recorded. All patients were prospectively followed for a median period of 3.4 years. The primary outcome of this study was a composite outcome of cardiovascular (CV) events (both fatal and nonfatal) or a doubling of serum creatine or progression to end-stage kidney disease (ESKD), whichever occurred first. Results: At baseline, the median age of patients was 65.2 years; 36.4% had diabetes; 21.4% had a history of CV disease; the mean of estimated glomerular filtration rate (eGFR) was 33 mL/min/1.73 m^2^; and the means of OBP, AOBP, and daytime ABPM were 151/84 mm Hg, 134/77 mm Hg, and 132/77 mm Hg, respectively. During the follow-up, 18 patients had a CV event, and 37 patients had a renal event. In the univariate cox regression analysis, systolic AOBP was found to be predictive of the primary outcome (HR per 1 mm Hg increase in BP, 1.019, 95% CI 1.003–1.035), and after adjustment for eGFR, smoking status, diabetes, and a history of CV disease and systolic and diastolic AOBP were also found to be predictive of the primary outcome (HR per 1 mm Hg increase in BP, 1.017, 95% CI 1.002–1.032 and 1.033, 95% CI 1.009–1.058, respectively). Conclusions: In patients with CKD, AOBP appears to be prognostic of CV risk or risk for kidney disease progression and could, therefore, be considered a reliable means for recording BP in the office setting.

## 1. Introduction

Chronic kidney disease (CKD) is a global health concern. The incidence of end-stage kidney disease (ESKD) keeps rising in many industrialized nations, regardless of the widespread use of treatments to limit the progression of CKD [1]. It has been documented that between 85% and 95% of individuals with CKD (stages 3–5) experience hypertension [2]. Among patients with CKD, as the glomerular filtration rate (GFR) declines from 85 to 15 mL/min per 1.73 m^2^, the hypertension prevalence increases from 65% to 95%, according to the Modification of Diet in Renal Disease study [3]. Hypertension is the principal risk factor causing death in CKD patients, accounting for 45% of male deaths and 46% of female deaths [4]. However, high blood pressure is one of the most significant risk factors for CKD that can be improved [5]. Consequently, effective diagnosis and treatment of hypertension to reach blood pressure (BP) targets in patients with CKD can affect CKD progression and lower the high risk of cardiovascular disease (CVD) [6].

Similarly to the general population, the diagnosis of hypertension in patients with CKD is based on office blood pressure measurements (OBP) and, in recent decades, ambulatory blood pressure measurements (ABPM). The latter method was found to be more accurate, not only in the identification of hypertension but also in the prediction of CVD risk [7], despite its restrictive costs. Automated office blood pressure monitoring (AOBP) has been proposed to increase the accuracy of OBP readings [8]. AOBP necessitates that readings be collected by using a fully automated oscillometric sphygmomanometer that takes repeated BP recordings with the patient sitting alone and in a quiet environment. In a study of hypertensive individuals with CKD, conventional OBP readings were considerably higher than AOBP readings, but there was no difference between AOBP measurements and 24 h ABPM [9].

AOBP correlates better than OBP with left ventricular mass [10] and microalbuminuria [11] in patients with hypertension and also with intima–media thickness of the carotid artery in normotensive individuals [12]. This correlation is similar to that of daytime ABPM. Nevertheless, information on the predictive utility of AOBP measurements in hypertensive individuals with CKD is practically nonexistent. This prospective cohort study aimed to examine the predictive role, principally of AOBP, and secondarily of OBP and ABPM, with regard to the occurrence of CV or renal events in a cohort of 140 patients with hypertension and CKD stages 3–5 who were not on dialysis.

## 2. Materials and Methods

### 2.1. Study Participants

The participants of this study were recruited at the nephrology clinics of two Greek hospitals, Athens General Hospital Evangelismos and University General Hospital of Alexandroupoli. The inclusion criteria were patients with hypertension who were under treatment, had CKD stage 3–5 (according to the CKD-EPI equation), were not on dialysis, and were between 18 and 85 years of age. Exclusion criteria included acute kidney injury within the previous 2 months; kidney transplantation; pregnancy; hemodialysis or peritoneal dialysis; a change in antihypertensive therapy during the previous 15 days; and serious non-cardiovascular diseases such as cancer or liver cirrhosis. Those unable to attend follow-up office visits were omitted as well.

All eligible individuals received a medical interview, anthropometric measurements, and blood and urine tests at the start of the study. Specifically, laboratory data (serum creatinine, serum fasting glucose, intact parathyroid hormone, cholesterol, tri-glycerides, high-density lipoprotein-cholesterol, low-density lipoprotein-cholesterol, intact parathormone hormone, and urine albumin-to-creatinine ratio in a first morning void urine sample) were collected during the initial study visit. Each participant’s history of smoking, hyperlipidemia, and risk factors for CVD had been assessed. Serum creatinine concentrations were measured, and the estimated glomerular filtration rate (eGFR) was obtained by using the CKD-EPI equation adjusted for body surface area. This study was conducted in conformity with the principles of the Declaration of Helsinki. All participants gave their informed consent, and the study was authorized by the hospitals’ ethical committees and scientific boards.

### 2.2. Blood Pressure Measurements

All BP readings (OBP, AOBP, and ABPM) were taken in hospital outpatient settings. OBP was computed as the mean of two consecutive measurements by using a certified digital oscillometric BP electronic instrument (Microlife BP A100; Microlife AG, Widnau, Switzerland) and a medium–large stable bladder, 22–42 cm. BP measurements were obtained at 1 min intervals from 8:30 to 10:30. Before assessing OBP, participants were instructed to stay seated for 5 min and to avoid speaking; a doctor from the research team was present throughout. Systolic blood pressure (SBP) lower than 140 mm Hg and diastolic blood pressure (DBP) lower than mm Hg were deemed acceptable levels for OBP [13]. After the OBP measurement, subjects received BP measurements by using ABPM and AOBP on the same and subsequent days, respectively.

For 24 h ABPM monitoring, a certified electronic Microlife WatchBP O3, bladder size 22–32 cm, or 32–42 cm, was employed. The device was set to take a reading every 15 min during the day (06:00–23:00) and every 30 min during the night (00:00–06:00) (23:00–06:00). At the time of measurement, participants were told to immobilize their arm and record their day activities and sleep duration in a notebook. SBP lower than 135 mm Hg and DBP lower than 85 mm Hg were accepted values of normality for ABPM in adults during daytime (awake); night-time (asleep) SBP and DBP lower than 120 and 70 mm Hg, respectively; and 24 h SBP and DBP lower than 130 and 80 mm Hg, respectively [13]. Awake SBP and DBP levels were determined by using the participants’ diaries detailing their waking and sleeping habits. Twenty valid daytime and seven valid nocturnal BP measurements were necessary to consider the test valid [14].

The Microlife WatchBP Office device, which was validated for OBP measurement by the International Protocol [15], was used to obtain AOBP readings. The subject was guided into the examination room and placed in a chair with arms supported by adjustable armrests at heart level and uncrossed feet on the floor. The person was left alone to rest for 5 min. After that, the device was turned on remotely from a desktop computer in the office next door by using Bluetooth to start measuring both arms at the same time. The device was programmed to record BP every minute (timed from the end of one reading to the start of the next). All six measurements were utilized to calculate the mean AOBP [16]. Normal values of AOBP were established as SBP lower than 135 mm Hg and DBP lower than 85 mm Hg [17].

### 2.3. Follow-Up

Beginning with the initial study visit, participants attended follow-up meetings every six months to monitor BP management, antihypertensive therapy, and primary outcomes. The primary outcome of this study was a composite outcome of a CV event, serum creatine doubling, or development of ESKD, whichever came first. CV events comprise fatal or nonfatal CV events, such as myocardial infarction; coronary heart disease, including coronary intervention, hospitalization for heart failure (HF), stroke, and intermittent claudication; and peripheral arterial disease interventions, whichever happened first. Renal events were defined as serum creatinine levels doubling or the progression to ESKD. The endpoint of ESKD was met on the first day of dialysis treatment.

### 2.4. Statistical Analysis

The Kolmogorov–Smirnov test was utilized to examine the data for normality. Normally distributed continuous variables are expressed as the mean standard deviation, non-normally distributed continuous variables are expressed as the median (interquartile range), and categorical variables are expressed as a percentage. Categorical variables were compared by using the χ^2^ test, and continuous variables were compared by using Student’s *t* test. We utilized Cox proportional hazard models to assess the relationship between the initial BP values (OBP, AOBP, and ABPM) and the onset of a CV or renal event. The correlation between OBP, AOBP, and ABPM values and the primary outcome was evaluated by using hazard ratios and their 95% confidence intervals per 1 mm Hg increase in systolic and diastolic BP. In the Cox regression analysis, two models were conducted: the first model was univariate-unadjusted, and the second model was multivariate-adjusted for baseline covariates known to influence renal and CV outcomes, such as smoking status (never, former, current), diabetes, eGFR, and a history of CVD. Moreover, by using the Cox regression analysis, we developed survival curves, which were adjusted for smoking status (never, former, current), diabetes, eGFR, and a history of CVD, in various groups of patients based on normal and high levels of BP assessed at baseline by using the three distinct techniques, OBP, AOBP, and ABPM. All *p* values are two-sided, and *p* < 0.05 denotes statistical significance. All statistical analyses were performed by using version 20.015 of the MedCalc statistical tool.

## 3. Results

### 3.1. Entry and Follow-Up Data

We included 140 of 161 eligible patients. The reasons for exclusion were inadequate ABPM recordings *(n* = 9), changes in antihypertensive treatment 2 weeks before the study (*n* = 8), and acute kidney injury on top of CKD in the last two months (*n* = 4). Six patients were lost to follow-up. The median age of the patients was 68 (57–76) years, and 50 (35.7%) were women. In total, 51 patients (36.4%) had diabetes, 30 patients (21.4%) had a history of CVD, 83 patients (59.3%) were receiving drugs for hyperlipidemia, and 19 patients (13.6%) were current smokers. The number of patients with CKD stages 3a, 3b, 4, and 5 was 39 (28%), 42 (30%), 47 (33.5%), and 12 (8.5%), respectively, with a mean eGFR of 33 ± 14 mL/min/1.73 m^2^. Demographic, clinical, and treatment information on the 140 patients included in the cohort is presented in Table 1.

At baseline, the mean systolic/diastolic OBP, AOBP, and daytime ABPM were 151/81 mm Hg, 134/77 mm Hg, and 132/77 mm Hg, respectively (Table 1). The mean differences between systolic AOBP/OBP and AOBP/daytime ABPM were 17 mm Hg and 2 mm Hg, respectively. The median follow-up time was 3.4 (2–4.5) years. A total of 18 patients had at least 1 CV event, and 37 patients had a renal event. CV events were as follows: 3 fatal acute myocardial infarctions, 1 fatal aortic dissection, 1 coronary intervention, 5 hospitalizations for HF, 6 acute myocardial infarctions, and 2 interventions for peripheral arterial disease. Renal events were as follows: 31 patients developed ESKD (29 on hemodialysis and 2 on peritoneal dialysis), and 6 had a doubling of serum levels of creatinine.

Compared with patients without an event, patients who experienced CV or renal events during the follow-up period had lower eGFRs; higher levels of iPTH and albuminuria; higher systolic daytime, night-time and 24 h ABPM values; and higher numbers of antihypertensive drugs used (Table 2).

### 3.2. Blood Pressure and Outcomes

From the Cox regression analysis, neither the systolic nor the diastolic OBP measurements were predictive of the primary outcome by using the unadjusted model; however, the diastolic OBP readings were found to be predictive of the primary outcome according to the adjusted model (HR per 1 mm Hg increase in BP 1.026, 95% CI 1.003–1.049). Progressing with the analysis, the systolic AOBP readings were predictive of CV or renal events by using the unadjusted and adjusted models (HR per 1 mm Hg increase in BP 1.019, 95% CI 1.003–1.035 and 1.017, 95% CI 1.002–1.032, respectively). The diastolic AOBP readings were not predictive of the primary outcome when fitted to the unadjusted model but were predictive according to the adjusted model (HR per 1 mm Hg increase in BP 1.033, 95% CI 1.009–1.058). Systolic daytime ABPM appeared to be predictive of the primary outcome by using the unadjusted model (HR per 1 mm Hg increase in BP, 1.024, 95% CI 1.007–1.040) but was not predictive with the adjusted model, and diastolic daytime ABPM was not found to be predictive by using the unadjusted model but was predictive with the adjusted model (HR per 1 mm Hg increase in BP, 1.033, 95% CI 1.004–1.063). Systolic night-time and 24 h ABPM were predictive according to the unadjusted model (HR per 1 mm Hg increase in BP 1.031, 95% CI 1.016–1.046 and 1.030, 95% CI 1.014–1.047, respectively) and the adjusted model (HR per 1 mm Hg increase in BP 1.024, 95% CI 1.009–1.039 and 1.022, 95% CI 1.006–1.039, respectively). On the other hand, diastolic night-time and 24 h ABPM were not predictive when fitted to the unadjusted model but were predictive according to the adjusted model (HR per 1 mm Hg increase in BP 1.041, 95% CI 1.013–1.069 and 1.042, 95% CI 1.012–1.073, respectively). All the results are presented in Table 3.

Figure 1 shows the survival curves, which are adjusted for smoking status (never, former, current), diabetes, eGFR, and a history of CVD, depicting the relationship between the incidence of CV or renal events and normal or high baseline systolic BP measured with OBP, AOBP, and ABPM. A marginally significant association between the incidence of CV or renal events and high systolic BP was observed with AOBP; meanwhile, elevated systolic night-time ABPM appeared to be highly prognostic (*p* = 0.050 and *p* = 0.003, respectively). On the other hand, high systolic OBP and daytime and 24 h ABPM were of no prognostic importance. Regarding normal and high baseline diastolic BP, which were measured via the three methods, fitting the adjusted survival curves, it emerged that only high diastolic night-time ABPM was related to the primary outcome of this study (*p* = 0.004) (Figure 2).

## 4. Discussion

With this prospective cohort study, we were the first to investigate the prognostic effect of AOBP measurements in hypertensive patients with stage 3 to 5 CKD who were not on dialysis. As a result, high systolic AOBP readings were found to be a significant factor for predicting the composite outcome of CV or renal events even after adjustment for smoking status, eGFR, diabetes, and a history of CVD. Furthermore, diastolic AOBP measurements were also found to be prognostic for CV or renal events when fitted to the adjusted model. Taken together, these results suggest that high systolic and diastolic AOBP readings are significant risk factors for the prognosis of hypertensive patients with CKD stage 3–5 in terms of CV and kidney disease.

These findings are partially in line with those of the study of Myers et al. (2015) in a non-CKD population in which the prognostic values of AOBP readings for the development of future CV events were compared [18]. Included in the Cardiovascular Health Awareness Program (CHAP) were 3627 community-dwelling adults older than 65 without hypertension treatment. At a community pharmacy, sitting and undisturbed individuals provided AOBP measurements. Participants were monitored for a mean of 4.9 years for CV-related fatal and nonfatal events. Adjusted hazard ratios (95% confidence intervals) were calculated for 10 mm Hg increases in BP (mm Hg) by using Cox proportional hazards regression, using the group with the smallest event rate as the reference group. A CV event was attended by 271 individuals in total. With an SBP between 135 and 144 mm Hg and DBP between 80 and 89 mm Hg, the hazard ratio increased significantly.

In a second study, which was similarly conducted in a non-CKD population (6.183 CHAP participants 66 years old on antihypertensive treatment at baseline), the association between obtained AOBP at baseline and CV events was investigated [19]. The average duration of follow-up was 4.6 years. Adjusted hazard ratios (with 95% confidence intervals) were calculated for 10 mm Hg increments in the achieved AOBP at baseline, utilizing Cox proportional hazards regression and the BP group with the lowest event rate as the reference group. According to 904 fatal and nonfatal CV events, the lowest group of SBP for CV events was 110 to 119 mm Hg, which was less than the next highest group of 120 to 129 mm Hg. Regarding the DBP, the hazard ratio over 60 mm Hg remained basically stable.

In the SPRINT study, 9361 patients with an SBP of 130 mm Hg or more and a high CV risk were randomly given an SBP goal of less than 120 mm Hg (intensive therapy) or less than 140 mm Hg (standard therapy) [20]. Myocardial infarction, other acute coronary syndromes, stroke, HF, or death from CV causes were included in the primary composite outcome. In the SPRINT study, the AOBP method was used, which requires the patient to be quiet and alone, while an electronic oscillometric sphygmomanometer takes several readings. This is the same method we used in our investigation. At one year, the intensive therapy group had a mean SBP of 121.4 mm Hg compared to 136.2 mm Hg for the conventional therapy group. Due to a considerably lower rate of the primary composite outcome in the intensive therapy group compared to the standard therapy group, the intervention was terminated early after a median follow-up of 3.26 years. In the intensive therapy group, death from all causes was also significantly reduced. These results are close to the findings of our study, in which higher systolic AOBP readings were related to worse CV and renal prognosis. In contrast, in a subgroup analysis among individuals with CKD at baseline (mean eGFR 48 mL/min per 1.73 m^2^) in the SPRINT trial, patients were randomly allocated to an SBP goal of 120 mm Hg (intense group; *n* = 1330) or 140 mm Hg (standard group; *n* = 1316) [21]; 112 individuals in the intensive group and 131 in the standard group experienced the primary composite CV outcome after a median follow-up of 3.3 years of follow-up. The mean SBP was 123 mm Hg in the intensive group and 135 mm Hg in the standard group throughout the course of the study. Following the analysis, the authors concluded that in patients with mild-to-moderate CKD and hypertension without diabetes, extensive reductions in SBP led to an important reduction in the incidence of CVD and all-cause mortality. These conclusions are also, to some extent, consistent with our findings about the elevated systolic AOBP readings in a relatively comparable CKD group.

In a comparison of OBP and AOBP, it has been demonstrated that AOBP reduces the white coat effect, prevents speaking during the rest and measurement periods, and eliminates observer error and bias [22,23,24]. In the CAMBO study (Conventional Versus Automated Blood Pressure Measurement in the Office) [25], AOBP and OBP were evaluated for hypertension management in everyday, community-based clinical practices. In this study, 88 primary care doctors in 67 practices across 5 locations in eastern Canada were randomly assigned to either utilize AOBP or continue using OBP. The difference between systolic daytime ABPM and systolic BP at the first return visit was substantially less in the AOBP group (2.3 mm Hg) than in the control OBP group (6.5 mm Hg). In addition, the association between AOBP and daytime ABPM was considerably stronger than the association between OBP and awake-daytime ABPM. AOBP showed a greater correlation with target organ damage, including intima–media thickness of the carotid artery and left ventricular mass index, compared to OBP [10,11,12].

In our study comparing the mean SBP measurements with the three different methods, namely, OBP, AOBP, and ABPM, we found that the mean difference in systolic BP between AOBP and OBP was 17 mm Hg, while the difference in mean SBP between AOBP and daytime ABPM was only 2 mm Hg. These results are in concordance with the results of a cross-sectional study that included 91 CKD patients (mean eGFR 45 mL/min per 1.73 m^2^) for whom BP measurements were obtained with the 3 different methods: OBP, AOBP, and ABPM [9]. The investigators concluded that in these CKD patients, OBP measurements were significantly higher than measurements obtained by using AOBP in a quiet room, but there was no significant difference in this setting between AOBP readings and 24 h ABPM values.

Regarding the prognostic role of the other two methods of BP measurement that were implemented in our study, systolic OBP was not predictive of the primary outcome, while diastolic OBP appeared to be prognostic of CV or renal events only when the adjusted model was applied. Systolic and diastolic ABPM were revealed to be prognostic of CV or renal events. These results are consistent with the findings of Minutolo et al. (2011), who investigated the prognostic role of daytime systolic and diastolic BP in comparison with OBP in 436 individuals with CKD (mean eGFR 42.9 mL/min per 1.73 m^2^) [7]. The progression of ESKD, or death, and the occurrence of fatal and nonfatal CV events were the primary outcomes. In total, 155 and 103 patients, respectively, attained the renal and CV end points during the follow-up (median, 4.2 years). Those with a daytime SBP of 136 to 146 mm Hg and those with a daytime SBP greater than 146 mm Hg had an elevated adjusted risk of the CV end point and renal death in comparison to those with a daytime systolic BP of 126 to 135 mm Hg. On the contrary, neither renal nor cardiovascular endpoint risk was predicted by OBP.

There are some limitations in the current study that need to be reported. First of all, the relatively small size of our cohort could limit the generalization of our findings. Furthermore, the order of the BP measurements (OBP, ABPM, and AOBP) in the current study was always the same; meanwhile, a random order could be a more accurate method to compare the three techniques. In addition, home BP measurements could be employed to compare in-office and out-of-office BP measurement techniques comprehensively. Finally, our findings are based on a single BP measurement with the three different methods in two consecutive days at the beginning of the study. However, repeated measurements over time and longitudinal change could provide more a precise ability to predict outcomes.

## 5. Conclusions

In conclusion, systolic and diastolic AOBP measurements perform well in the prediction of CV or renal risk in CKD hypertensive patients not on dialysis. The findings of this study affirm the role of AOBP as the reference technique for the assessment of BP in the office setting in patients with CKD. These results are in line with the KDIGO 2021 Clinical Practice Guideline for the Management of Blood Pressure in Chronic Kidney Disease [26], which proposes AOBP as the preferred technique to measure BP in the office setting.

## Figures and Tables

**Figure 1 healthcare-11-01360-f001:**
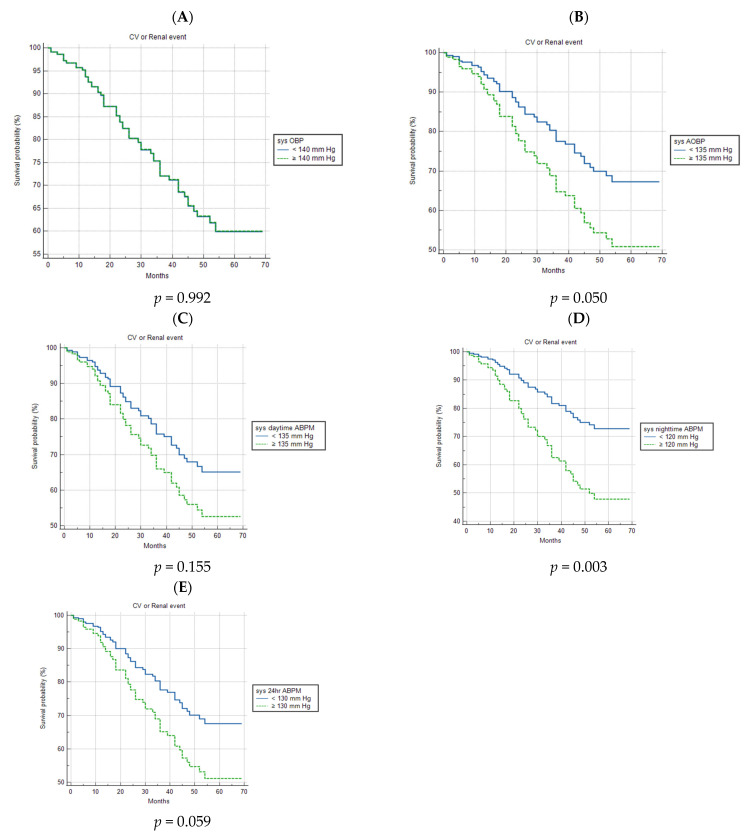
Survival curves, which are adjusted for smoking status (never, former, current), diabetes, eGFR, and a history of CVD, as a function of patients with normal or high systolic blood pressure at baseline, as measured with OBP, AOBP, and ABPM. (**A**) Survival curves for CV or renal events in patients with normal and high systolic OBP. (**B**) Survival curves for CV or renal events in patients with normal and high systolic AOBP. (**C**) Survival curves for CV or renal events in patients with normal and high systolic daytime ABPM. (**D**) Survival curves for CV or renal events in patients with normal and high systolic night-time ABPM. (**E**) Survival curves for CV or renal events in patients with normal and high systolic 24 h ABPM. ABPM—ambulatory blood pressure measurement; AOBP—automated office blood pressure; CV—cardiovascular; OBP—office blood pressure; sys—systolic.

**Figure 2 healthcare-11-01360-f002:**
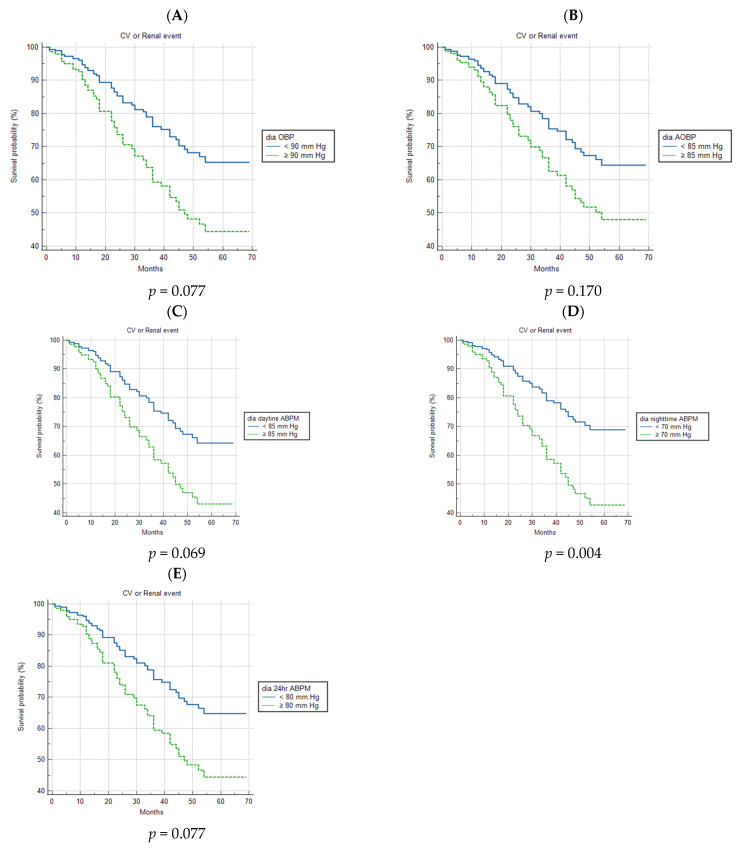
Survival curves, which are adjusted for smoking status (never, former, current), diabetes, eGFR, and a history of CVD, as a function of patients with normal or high diastolic blood pressure at baseline, as measured with OBP, AOBP, and ABPM. (**A**) Survival curves for CV or renal events in patients with normal and high diastolic OBP. (**B**) Survival curves for CV or renal events in patients with normal and high diastolic AOBP. (**C**) Survival curves for CV or renal events in patients with normal and high diastolic daytime ABPM. (**D**) Survival curves for CV or renal events in patients with normal and high diastolic night-time ABPM. (**E**) Survival curves for CV or renal events in patients with normal and high diastolic 24 h ABPM. ABPM—ambulatory blood pressure measurement; AOBP—automated office blood pressure; CV—cardiovascular; OBP—office blood pressure; sys—systolic.

**Table 1 healthcare-11-01360-t001:** Baseline characteristics of the study population.

Women (N, %)	50 (35.7)
Age (y)	68 (57–76)
BMI (kg/m^2^)	27.8 (25.5–31.9)
eGFR (mL/min/1.73 m^2^)	33 ± 14
CKD stages	
3a (N, %)	39 (28)
3b (N, %)	42 (30)
4 (N, %)	47 (33.5)
5 (N, %)	12 (8.5)
History of CV disease (N, %)	30 (21.4)
Statin use—hyperlipidemia (N, %)	83 (59.3)
Current smoker (N, %)	19 (13.6)
Diabetes (N, %)	51 (36.4)
Number of antihypertensive drugs used	2.3 ± 1.2
RAS blockade (N, %)	93 (66)
iPTH (pg/mL)	102 (74–156)
Albumin-to-creatinine ratio (mg/g)	166 (33–641)
Blood pressure (mm Hg)	
Systolic OBP	151 ± 21
Diastolic OBP	84 ± 12
Systolic AOBP	134 ± 17
Diastolic AOBP	77 ± 12
Systolic daytime ABPM	132 ± 16
Diastolic daytime ABPM	77 ± 11
Systolic night-time ABPM	122 ± 18
Diastolic night-time ABPM	68 ± 11
Systolic 24 h ABPM	129 ± 16
Diastolic 24 h ABPM	77 ± 11

ABPM—ambulatory blood pressure measurement; AOBP—automated office blood pressure; BMI—body mass index; CKD—chronic kidney disease; eGFR—estimated glomerular filtration rate; OBP—office blood pressure; RAS—renin–aldosterone system. Data are presented as the mean ± SD or median (interquartile range) for continuous variables and as a percentage for categorical variables.

**Table 2 healthcare-11-01360-t002:** Baseline characteristics of the two groups of patients based on whether they experienced CV or renal events during the follow-up period of the study.

Variables	Patients Who ExperiencedCV or Renal Events		
	No	Yes	*p* Value
Ν	79	55	
Women (N, %)	25 (32)	21(38)	0.06
Age (y)	68 (56–76)	68 (62–74)	0.50
BMI (kg/m^2^)	28 (25.5–32)	28 (26.5–32.4)	0.31
Baseline eGFR (mL/min/1.73 m^2^)	39 ± 12	25 ± 13	0.00
CKD stages			
3a (N, %)	31 (39)	7 (13)	0
3b (N, %)	30 (38)	10 (19)	0.02
4 (N, %)	17 (22)	27 (49)	0
5 (N, %)	1 (1)	11 (20)	0
Diabetes (N, %)	25 (32)	24 (44)	0.16
Number of antihypertensive drugs used	1.9 ± 1	2.7 ± 1.2	0.00
RAS blockade (N, %)	57 (72)	35 (64)	0.09
iPTH (pg/mL)	94 (60–116)	152 (99–230)	0.00
Albumin-to-creatinine ratio (mg/g)	64 (19–309)	546 (161–1345)	0.00
Blood pressure (mm Hg)			
Systolic OBP	149 ± 19	153 ± 22	0.20
Diastolic OBP	83 ± 11	84 ± 14	0.54
Systolic AOBP	131 ± 15	136 ± 19	0.13
Diastolic AOBP	76 ± 11	78 ± 13	0.57
Systolic daytime ABPM	129 ± 15	135 ± 16	0.02
Diastolic daytime ABPM	76 ± 11	78 ± 12	0.35
Systolic night-time ABPM	118 ± 15	128 ± 19	0.00
Diastolic night-time ABPM	66 ± 11	69 ± 11	0.18
Systolic 24 h ABPM	125 ± 14	133 ± 17	0.00
Diastolic 24 h ABPM	73 ± 10	75 ± 11	0.23

ABPM—ambulatory blood pressure measurement; AOBP—automated office blood pressure; BMI—body mass index; CKD—chronic kidney disease; eGFR—estimated glomerular filtration rate; OBP—office blood pressure; RAS—renin–aldosterone system. Data are presented as the mean ± SD or median (interquartile range) for continuous variables and a percentage for categorical variables.

**Table 3 healthcare-11-01360-t003:** Hazard ratios (HRs) with 95% confidence intervals (CIs) and *p* values for the composite outcome of CV or renal events in relation to OBP, AOBP, and ABPM (unadjusted and adjusted model). Multivariate model was adjusted for eGFR (mL/min per 1.73 m^2^), smoking status (never, former, current), history of CV disease, and diabetes.

BP	Unadjusted-Univariate Model		Adjusted-Multivariate Model	
Systolic (mm Hg)	HR (95% CI)	*p*	HR (95% CI)	*p*
OBP	1.010 (0.997–1.022)	0.124	1.009 (0.996–1.022)	0.154
AOBP	1.019 (1.003–1.035)	0.017	1.017 (1.002–1.032)	0.028
Daytime ABPM	1.024 (1.007–1.040)	0.005	1.016 (0.998–1.033)	0.068
Night-time ABPM	1.031 (1.016–1.046)	0.000	1.024 (1.009–1.039)	0.002
24 h ABPM	1.030 (1.014–1.047)	0.000	1.022 (1.006–1.039)	0.009
Diastolic (mm Hg)	HR (95% CI)	*p*	HR (95% CI)	*p*
OBP	1.009 (0.988–1.030)	0.402	1.026 (1.003–1.049)	0.026
AOBP	1.014 (0.993–1.036)	0.206	1.033 (1.009–1.058)	0.006
Daytime ABPM	1.014 (0.992–1.036)	0.204	1.033 (1.004–1.063)	0.023
Night-time ABPM	1.021 (0.997–1.046)	0.083	1.041 (1.013–1.069)	0.004
24 h ABPM	1.020 (0.996–1.043)	0.110	1.042 (1.012–1.073)	0.006

ABPM—ambulatory blood pressure measurement; AOBP—automated office blood pressure; BP—blood pressure; CV—cardiovascular; HR—hazard ratio; OBP—office blood pressure.

## Data Availability

Data are available on request due to ethical restrictions.

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
