# Peer review of "The Prognostic Role of Automated Office Blood Pressure Measurement in Hypertensive Patients with Chronic Kidney Disease"

_healthcare, 2023, doi:10.3390/healthcare11101360_

Round 1
Reviewer 1 Report
The authors describe the prognostic value of automated office BP, office BP and ambulatory BP readings in CKD patients in relation to cardiovascular and renal endpoints.
It is a well-written manuscript with interesting results and a clear conclusion. I have some comments and suggestions for improvements.
1) You describe how office BP is measured. Why only use 2 measurements and not 3 and the average of the 2 last measurements ?
2) Do you know the difference between the first and second office BP ?
3) What does it mean when you say "a doctor from the research team was present throughout" ? did the doctor measure the BP and did he/she sit in front of the patient ?
4) ABPM was performed just after the office BP recording, but how long time did it go before the AOBP was performed ? did you measure a new office BP on the day of the AOBP ?
5) Follow-up: How did you manage deaths not related to CV disease ?
6) Is it possible to depict the relations: a) between OBP and AOBP and b) between day-time ABPM and AOBP using linear associations or Bland-Altmann plots to better illustrate how these measurements fit together.
7) Table 2: are the terms adjusted and unadjusted correctly placed. It is best to show unadjusted data first (left column) and adjusted next (right hand column).
8) You should include information about albuminuria (or proteinuria) and adjust for this parameter.
9) You mention the patients have their BP measured each 6. months during follow-up. Is this OBP or AOBP ? if antihypertensive treatment during follow-up is adjusted according to one or the other how does this affect their prognostic value ?
Reviewer 2 Report
This study evaluates the prognostic role of automated office blood pressure measurement in hypertensive patients with chronic kidney disease.
I have read the manuscript carefully and have to say there is a problem with scientific soundness. I don't understand how "automated office blood pressure measurement" or any other technique of blood pressure (BP) measurement by itself may have the prognostic role. Different methods of BP measurement may differ in their sensitivity for assessing changes in BP value, but they can’t have the "prognostic" role of CV risk and kidney disease progression.
On the other hand, novelty of the study is low. Hypertension is well established risk factor of CV risk and chronic kidney disease progression, and patients with CKD are routinely monitored for hypertension.
Author Response
Please see the response file. (attached)

Reviewer 3 Report
Authors followed their CKD patients with hypertension and found that AOBP measurement is prognostic of CV and renal outcomes. Comments are as the followings.
1. Every clinical practitioner knows that office BP is not representative, and other methods have to be considered. Did authors ever consider comparing the value of self-monitored BP at home with either of these methods in this study? Asking the patients to measure their own BP at home would be another way to promote their own self-care.
2. This study did not seriously consider the influence of medications. Your adjustment models did not include medications. Also, the medications record in Table 1 only mentioned RAS blockade.
3. Is there any reason why in the top of table1, "Women (N %)" has to be bold?
4. Authors may consider making another table to show the difference of parameters between patients with or without events.
Author Response

(The authors gave the same response as above.)

Round 2
Reviewer 2 Report
I accepted authors response.
Just check Table 2. According to the values of eGFR, iPTH, albuminuria... you commented in the results, I think you made a technical mistake in Table 2 (you changed places of groups with experienced CV or renal event and without CV or renal event), rotate „Yes“ and „No“.
Reviewer 3 Report
My questions stay the same. Blood pressure measurement at office, without considering measurement at home, is questionable in term of reliability.
